# Association of perioperative serum carcinoembryonic antigen level and recurrence in low-risk stage IIA colon cancer

Han-Gil Kim[¤a], Seung Yoon Yang, Yoon Dae Han, Min Soo Cho, Byung Soh Min, Kang Young Lee, Nam Kyu Kim, Hyuk Hur[¤b]*

Department of Surgery, Severance Hospital, Yonsei University College of Medicine, Seoul, South Korea

¤a Current address: Department of Surgery, Gyeongsang National University Hospital, Gyeongsang National University College of Medicine, Jinju, Gyeongsangnam-do, South Korea
¤b Current address: Department of Surgery, Yongin Severance Hospital, Yonsei University College of Medicine, Seoul, South Korea
* HHYUK@yuhs.ac

## Abstract

### Background

The purpose is to investigate prognosis according to serum CEA levels before and after surgery in patients with stage IIA colon cancer who do not show high-risk features.

### Methods

Among the patients diagnosed with colon adenocarcinoma between April 2011 and December 2017, 462 patients were confirmed as low-risk stage IIA after surgery and enrolled. The ROC curve was used to determine cut-off values of pre- and postoperative CEA. Patients were classified into three groups using these new cut-off values.

### Results

All recurrence occurred in 52 of 463 patients (11.2%). However, recurrence in group H was 15.9%, which was slightly higher than the other two groups (P = 0.04). Group L and M showed 10.5% and 12.8% overall survival, group H was higher at 21.0% (P = 0.005). Recurrence was the only risk factor in group H was significantly higher in group L (HR 2.008, 95% CI, 1.123–3.589, P = 0.019). Mortality was similar to recurrence (HR 1.975, 95% CI 1.091–3.523, P = 0.044).

### Conclusion

Among patients with low-risk stage IIA colon cancer, recurrence and mortality rates were higher when perioperative serum CEA levels were above a certain level. Therefore, high CEA level should be considered a high-risk feature and adjuvant chemotherapy should be performed.

**Data Availability Statement:** All relevant data are within the paper and its Supporting information files.

**Funding:** The authors received no specific funding for this work.

**Competing interests:** The authors have declared that no competing interests exist.

## Introduction

Colorectal cancer (CRC), the third most commonly diagnosed cancer and the second leading cause of cancer death worldwide, has steadily increasing mortality rates; it is estimated that >1,800,000 new cases of CRC occurred and that >860,000 people died of this disease in 2018 [1]. Surgery is the main treatment for early cases, but patients are often diagnosed in the advanced stages and there also may be distant metastases [2]. Standard treatment of CRC without metastasis (stage I-III) is radical resection of cancer lesions. Among patients with node-positive CRC (stage III), it is common to perform postoperative adjuvant chemotherapy, and the effect of this treatment has already been demonstrated [3].

Colon cancer and rectal cancer differ somewhat in treatment. According to the National Comprehensive Cancer Network (NCCN) guidelines, postoperative observation is usually performed when stage IIA colon cancer is not high risk and adjuvant chemotherapy is considered an option. In rectal cancer, on the other hand, chemotherapy is the main treatment and observation is an option [4].

Carcinoembryonic antigen (CEA) is a tumor marker used to help manage colon cancer. CEA is used to guide cancer surveillance after surgery, and high pre- and postoperative CEA levels are as independent predictors of overall and disease-free survival [5]. The most widely used upper margin of the normal range of CEA concentration is 5 ng/ml [6]. In early stage colon cancer, however, CEA concentration is usually less than 5 ng/ml, which reduces diagnostic value. Colon cancer screening reduces mortality by identifying cancers at an earlier and more treatable stage [7]. As screening becomes popular, early stage colon cancer is increasing.

The purpose of this study is to present a difference in recurrence according to CEA concentrations in patients with stage IIA colon cancer who are not at high risk. Based on these results, we wanted to find out whether adjuvant chemotherapy could be added to a specific group.

## Materials and methods

We conducted a retrospective chart review of a prospectively maintained database of all patients who underwent curative resection of primary stage II colon cancer between January 2008 and December 2015. TNM pathologic stage II disease was diagnosed according to the American Joint Committee on Cancer (AJCC) Cancer Staging Manual 7th edition [8].

The exclusion criteria were T4 cancer, poorly differentiated and mucinous tumor, bowel obstruction or perforation, lympho-vascular invasion, perineural invasion, positive margins, number of lymph nodes analyzed after surgery <12, preoperative chemotherapy or radiotherapy, adjuvant chemotherapy, palliative resection, and lack of preoperative and postoperative CEA data. Data on patient demographics, perioperative clinical outcomes, pathologic outcomes, and disease status at last follow-up were collected from the database, and the electronic medical records were reviewed.

Of the total 1,682 stage II colon cancer patients, only 463 patients were enrolled in this study after the exclusion criteria were applied, and the data was analyzed from October 2019 to February 2020. Prior to access, all data was anonymized, and this study was approved by the Institutional Review Board of Yonsei University Severance Hospital and the informed consent was waived (IRB No. 4-2019-1242).

Preoperative CEA was defined as the CEA value closest to the time of surgery, and postoperative CEA was defined as the last CEA value within 1 month after surgery. The ROC curve revealed that the preoperative CEA cutoff point was 3.305 ng/mL, and the calculated AUC was 0.60 (95% CI, 0.53–0.67, P = 0.009). With a CEA cut-off point of 3.305 ng/mL, the sensitivity and specificity for predicting recurrence were 59.7% and 58.1%, respectively. The ROC curve revealed that the postoperative CEA cut-off point was 1.86 ng/mL, and the calculated AUC

was 0.61 (95% CI, 0.54–0.69, P = 0.003). With a CEA cut-off point of 1.86 ng/mL, the sensitivity and specificity for predicting recurrence were 54.5% and 64.0%, respectively.

Patients were grouped by CEA status as follows: (1) patients with low (<3.305 ng/mL) preoperative CEA and low (≤1.86 ng/mL) postoperative CEA (group L); (2) patients with elevated (≥3.305 ng/mL) preoperative CEA and low (<1.86 ng/mL) postoperative CEA or patients with low (<3.305 ng/mL) preoperative CEA and elevated (≥1.86 ng/mL) postoperative CEA (group M); and (3) patients whose preoperative and postoperative CEA levels were both elevated (group H).

Statistical analyses were performed using IBM SPSS Statistics ver. 25.0 (IBM Co., Armonk, NY, USA). To assess a cut-off value for CEA with the maximum Youden index, receiver operating characteristic (ROC) curve and area under the curve (AUC) calculations were performed. Recurrence-free survival and overall survival were estimated by the Kaplan-Meier method, and univariate analyses of the significance of prognostic factors were evaluated by the log-rank test. Hazard ratios (HRs) and 95% CIs were estimated using Cox regression models. A multivariate analysis of factors associated with recurrence rate was performed using the Cox proportional hazards model with the backward stepwise (likelihood ratio) method. Variables with P values of less than 0.1 on univariate analysis were included in the final multivariable model. P values <0.05 were considered statistically significant.

## Results

The patient demographics are shown in Table 1. The mean age was 64 years in group L, which was slightly lower than 70.1 years in group M and 68.9 years in group H. Gender, BMI, history of smoking, alcohol use and ASA scores were not statistically significant for each group. The overall mean preoperative CEA concentration was 5.18 mg/dl, 1.75 mg/dl for group L, 4.63mg/dl for group M, and 10.44 mg/dl for group H. The overall mean postoperative CEA was 2.17 mg/dl: 1.00 mg/dl for group L, 1.78 mg/dl for group M, and 4.12 mg/dl for group H.

Underlying disease was divided into six categories: hypertension, diabetes mellitus, liver disease, lung disease, heart disease and kidney disease. Among the 463 patients, 320 patients (69.1%) had underlying disease: 63.9% in group L, 67.9% in group M, 77.1% in group H. This increasing trend was statistically significant with a P value of 0.007. Diabetes mellitus was 31.8% higher in group H than in groups L and M (P = 0.005). The other underlying diseases did not show statistically significant results (Table 2).

Table 3 shows the perioperative outcomes, and although the P values were lower than 0.05, there was no clear trend in each group. Table 4 describes postoperative outcomes. Postoperative complication occurred in 24 out of 463 patients (5.2%), and there was no statistical significance between groups. Among the pathologic outcomes of cancer, differentiation was also not statistically significant. More than 27 lymph nodes were harvested in group L and group M, but only 23.13 were harvested in group H, which was statistically significant (P = 0.002). Recurrence occurred in 52 of 463 patients (11.2%). There was no significant difference between group L (8.8%) and group M (9.0%). However, recurrence in group H was 15.9%, which was higher than the other two groups with a p value of 0.04. Overall survival was similar to disease-free survival: group H (21.0%) was higher than group L (10.5%) and group M (12.8%), which was statistically significant (P = 0.005).

Disease-free survival and overall survival between groups are shown in Fig 1. In the case of disease-free survival, group H showed statistically significantly lower results than the other two groups (versus group L; P = 0.009, group M; P = 0.032). Overall survival was statistically significant with P value of 0.023 between group L and group H only.

**Table 1. Demographics of the patients with low risk stage IIA colorectal cancer patients.**

| | Total (n = 463) | Group L (n = 228) | Group M (n = 78) | Group H (n = 157) | P |
|---|---|---|---|---|---|
| Age (yrs) | | | | | |
| Mean (range) | 66.7 (30–94) | 64.0 (30–92) | 70.1 (42–86) | 68.9 (38–94) | <0.001 |
| <70 | 247 (53.3%) | 146 (64.0%) | 30 (38.5%) | 71 (45.2%) | <0.001 |
| ≥70 | 216 (46.7%) | 82 (36.0%) | 48 (61.5%) | 86 (54.8%) | |
| Gender, n(%) | | | | | |
| Male | 272 (58.7%) | 131 (57.5%) | 45 (57.7%) | 96 (61.1%) | 0.496 |
| Female | 191 (41.3%) | 97 (42.5%) | 33 (42.3%) | 61 (38.9%) | |
| Body mass index (kg/m$^2$) | | | | | |
| Mean (range) | 23.2 (13.0–41.2) | 23.4 (14.4–41.2) | 23.2 (15.5–29.4) | 23.0 (13.0–36.9) | 0.587 |
| <25 | 341 (73.7%) | 169 (74.1%) | 57 (73.1%) | 115 (73.2%) | 0.861 |
| ≥25 | 122 (26.3%) | 59 (25.9%) | 21 (26.9%) | 42 (26.8%) | |
| ASA score | | | | | |
| 1 | 142 (33.1%) | 70 (30.7%) | 20 (25.6%) | 52 (33.1%) | 0.715 |
| 2 | 209 (45.1%) | 108 (47.4%) | 38 (48.7%) | 63 (40.1%) | |
| 3 | 103 (22.2%) | 49 (20.2%) | 18 (23.1%) | 39 (24.8%) | |
| 4 | 9 (1.9%) | 4 (1.8%) | 2 (2.6%) | 3 (1.9%) | |
| PreCEA (mg/dl) | | | | | |
| Mean (range) | 5.18 (0.31–60.48) | 1.75 (0.31–3.30) | 4.63 (1.71–15.79) | 10.44 (3.31–60.48) | <0.001 |
| <5 | 349 (75.4%) | | | | |
| ≥5 | 114 (24.6%) | | | | |
| <3.3 | 258 (55.7%) | | | | |
| ≥3.3 | 205 (44.3%) | | | | |
| PostCEA (mg/dl) | | | | | |
| Mean (range) | 2.17 (0.22–41.45) | 1.00 (0.22–1.80) | 1.78 (0.82–6.52) | 4.12 (1.82–41.45) | <0.001 |
| <5 | 437 (94.4%) | | | | |
| ≥5 | 26 (5.6%) | | | | |
| <1.8 | 276 (59.6%) | | | | |
| ≥1.8 | 187 (40.4%) | | | | |
| History of smoking | 132 (28.5%) | 63 (27.6%) | 19 (24.4%) | 50 (31.8%) | 0.467 |
| History of drinking alcohol | 165 (35.6%) | 83 (36.4%) | 26 (33.3%) | 56 (35.7%) | 0.899 |

ASA = American Society of Anesthesiologists; preCEA: preoperative carcinoembryonic antigen; postCEA: postperative carcinoembryonic antigen

**Table 2. Underlying disease.**

| | Total (n = 463) | Group L (n = 228) | Group M (n = 78) | Group H (n = 157) | P |
|---|---|---|---|---|---|
| Underlying disease | | | | | |
| No | 143 (30.9%) | 82 (36.0%) | 25 (32.1%) | 36 (22.9%) | 0.007 |
| Yes | 320 (69.1%) | 146 (63.9%) | 53 (67.9%) | 121 (77.1%) | |
| Hypertension | 216 (46.7%) | 97 (42.5%) | 39 (50.0%) | 80 (51.0%) | 0.098 |
| Diabetes mellitus | 106 (22.9%) | 43 (18.9%) | 13 (16.7%) | 50 (31.8%) | 0.005 |
| Liver disease | 19 (4.1%) | 10 (4.4%) | 2 (2.6%) | 7 (4.5%) | 1.000 |
| Lung disease | 32 (6.9%) | 13 (5.7%) | 7 (9.0%) | 12 (7.6%) | 0.478 |
| Heart disease | 30 (6.5%) | 15 (6.6%) | 3 (3.8%) | 12 (7.6%) | 0.755 |
| Kidney disease | 11 (2.4%) | 3 (1.3%) | 1 (1.3%) | 7 (4.5%) | 0.061 |

**Table 3. Perioperative outcomes.**

| | Total (n = 463) | Group L (n = 228) | Group M (n = 78) | Group H (n = 157) | P |
|---|---|---|---|---|---|
| Tumor location | | | | | |
| Right sided colon | 217 (46.9%) | 113 (49.6%) | 47 (60.3%) | 57 (36.3%) | 0.020 |
| Left sided colon | 246 (53.1%) | 115 (50.4%) | 31 (39.7%) | 100 (63.7%) | |
| OP type | | | | | |
| Right hemicolectomy | 210 (45.4%) | 108 (47.4%) | 47 (60.3%) | 55 (35.0%) | 0.017 |
| Transverse colectomy | 5 (1.1%) | 3 (1.3%) | 0 (0%) | 2 (1.3%) | |
| Left hemicolectomy | 38 (8.2%) | 23 (10.1%) | 5 (6.4%) | 38 (8.2%) | |
| Anterior resection | 157 (33.9%) | 68 (29.8%) | 21 (26.9%) | 68 (43.3%) | |
| Low anterior resection | 50 (10.8%) | 24 (10.5%) | 5 (6.4%) | 21 (13.4%) | |
| Subtotal colectomy | 3 (0.6%) | 2 (0.9%) | 0 (0%) | 1 (0.6%) | |

Tables 5 and 6 show risk factors through a uni- and multivariate analyses of recurrence and mortality, respectively. Recurrence rate between group L and Group H was the only risk factor (HR 2.02, 95% CI, 1.13–3.67, P = 0.019).

Mortality was similar to recurrence. Mortality was higher in group H than in group L and this was statistically significant (HR 1.97, 95% CI, 1.09–3.15, P = 0.041). The difference, however, is that age and gender is a risk factor for mortality. Mortality tended to be much higher in people over 70 years of age (HR 4.44, 95% CI, 2.30–9.01, P<0.001). In addition, it was not significant in univariate analysis, but after multivariate analysis, women showed lower mortality than men (HR 0.46, 95% CI, 0.24–0.86, P<0.017).

Tables 7 and 8 show the results of subgroup analysis of patients with non-diabetic patients because there were more diabetic patients in group H than other groups. In the results of patients without diabetes, there were statistically significant differences in Groups L and H (HR 2.25, 95% CI, 1.13–4.51, P = 0.021), and statistically significant factors were not found in patients with diabetes.

## Discussion

Carcinoembryonic antigen (CEA) is a glycoprotein with increased serum levels during cancer progression. This is useful for diagnosing various cancers and also plays an important role in

**Table 4. Postoperative outcomes.**

| | Total (n = 463) | Group L (n = 228) | Group M (n = 78) | Group H (n = 157) | P |
|---|---|---|---|---|---|
| Postoperative Complication | 24 (5.2%) | 11 (4.8%) | 7 (9.0%) | 6 (3.8%) | 0.968 |
| Intestinal obstruction | 13 (2.6%) | 7 (3.1%) | 3 (3.8%) | 1 (0.6%) | |
| Urinary problem | 2 (0.4%) | 0 (0%) | 0 (0%) | 1 (0.6%) | |
| Anastomosis leakage | 11 (2.2%) | 2 (0.9%) | 2 (2.6%) | 4 (2.5%) | |
| Bleeding | 1 (0.2%) | 1 (0.4%) | 0 (0%) | 0 (0%) | |
| Wound infection | 2 (0.4%) | 0 (0%) | 2 (2.6%) | 0 (0%) | |
| Intra-abdominal abscess | 1 (0.2%) | 1 (0.4%) | 0 (0%) | 0 (0%) | |
| Differentiation | | | | | |
| Well | 51 (11.0%) | 27 (11.8%) | 11 (14.1%) | 13 (8.3%) | 0.324 |
| Moderate | 412 (89.0%) | 201 (88.2%) | 67 (85.9%) | 144 (91.7%) | |
| Harvested lymph nodes (n) | | | | | |
| Mean (range) | 25.92(12–113) | 27.15(12–81) | 27.95(12–113) | 23.13(12–72) | 0.002 |
| Recurrence | 52 (11.2%) | 20 (8.8%) | 7 (9.0%) | 25 (15.9%) | 0.040 |
| Expired | 67 (14.5%) | 24 (10.5%) | 10 (12.8%) | 33 (21.0%) | 0.005 |

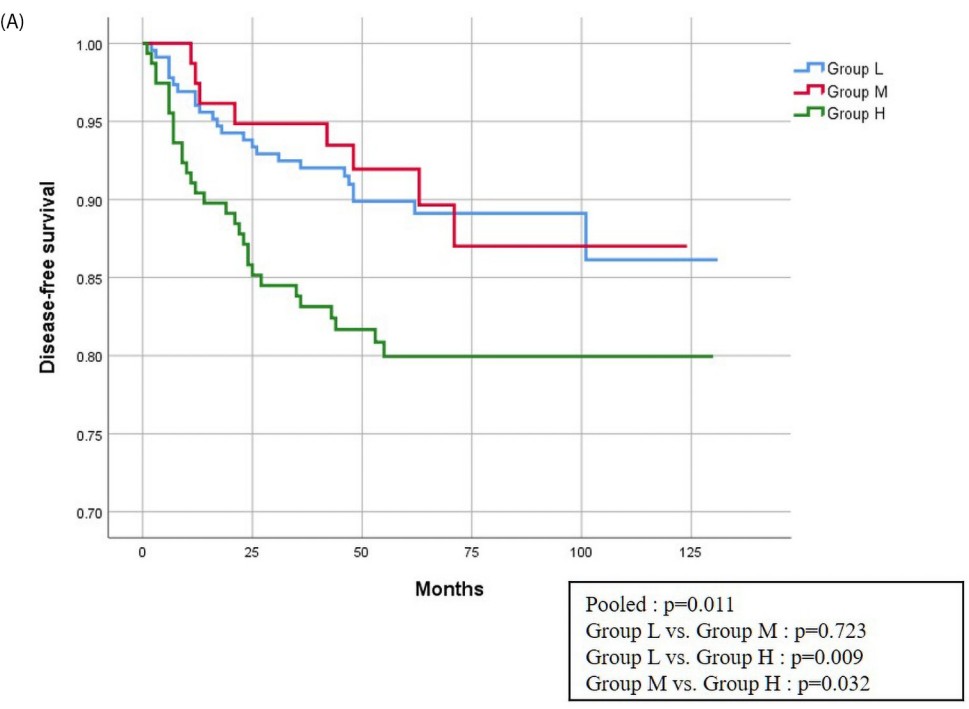

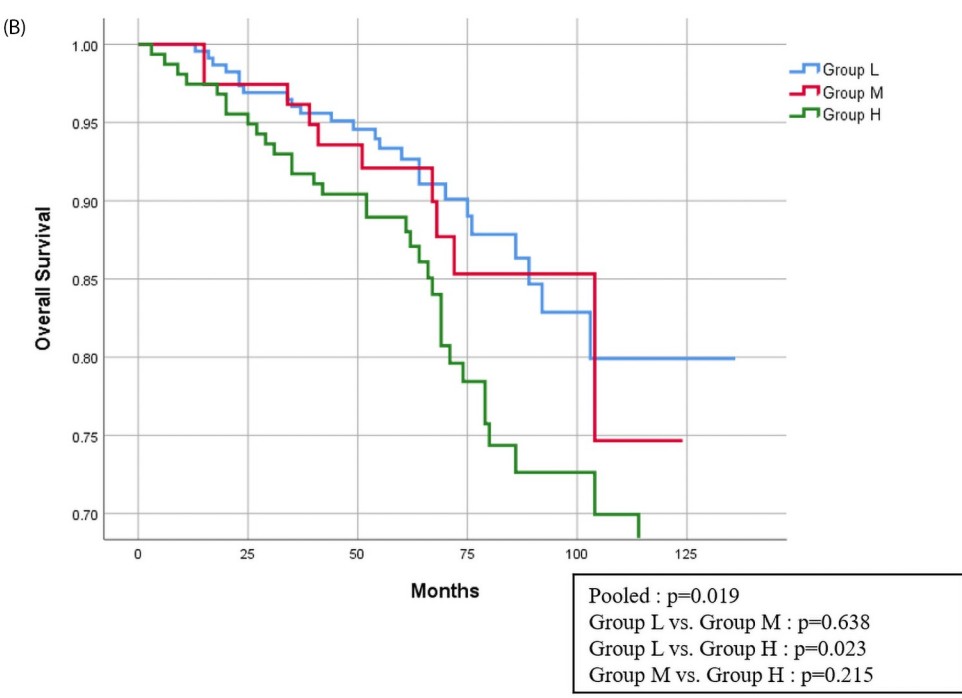

**Fig 1. Disease-free survival and overall survival between groups.**

**Table 5. Uni- and multivariate analysis of risk factors associated with recurrence using Cox regression model.**

| Factors | Univariate | Multivariate | | | |
|---|---|---|---|---|---|
| | *P* | HR | 95% CI | *P* | |
| Age (<70 vs. ≥70 years) | 0.572 | | | | |
| Gender (female vs. Male) | 0.476 | | | | |
| BMI (<25 vs. ≥25 kg/m2) | 0.258 | | | | |
| preCEA (<5 vs. ≥5 ng/mL) | 0.239 | | | | |
| postCEA (<5 vs. ≥5 ng/mL) | 0.759 | | | | |
| Group (L vs.H) | 0.012 | 2.02 | 1.13–3.67 | 0.019 | |
| Tumor site (Right vs. Left) | 0.100 | 1.55 | 0.85–2.91 | 0.165 | |
| Underlying disease | 0.131 | | | | |
| Smoking | 0.420 | | | | |
| Alcohol | 0.978 | | | | |
| Complication | 0.094 | 2.01 | 0.54–6.07 | 0.247 | |
| Histology (WD vs. MD) | 0.941 | | | | |

preCEA: preoperative carcinoembryonic antigen, WD: well differentiated, MD: moderately differentiated, HR: hazard ratio, 95% CI: 95% confidence interval

predicting recurrence surgical/medical treatment of cancer [9]. It is most relevant for colorectal cancer, but can also be seen in malignant tumors of the esophagus, stomach, liver, and pancreas [10, 11]. CEA is most affected by the pathologic TNM stage and is high even in the presence of lymphatic metastasis or nerve infiltration [12]. However, the low-risk stage IIA colon cancer that is the subject of this study rarely has a high CEA levels. This is more often less than 5 ng/dl, which is a criterion that is meaningful for efforts to begin raising serum CEA. The question of whether CEA level in low-risk stage IIA colon cancer overlooked simply because it is often lower than the reference value was the reason for this study.

The first purpose of this study was to analyze whether there was a relationship between perioperative CEA ratio and recurrence or mortality through a pilot study, but no statistical significance was found. In addition, based on the well-known CEA reference value of 5 ng/dl, we also investigated whether preoperative and postoperative CEA levels can serve as risk

**Table 6. Uni- and multivariate analysis of risk factors associated with mortality using Cox regression model.**

| Factors | Univariate | Multivariate | | | |
|---|---|---|---|---|---|
| | *P* | HR | 95% CI | *P* | |
| Age (<70 vs. ≥70 years) | <0.001 | 4.44 | 2.30–9.01 | <0.001 | |
| Gender (female vs. Male) | 0.077 | 0.46 | 0.24–0.86 | <0.017 | |
| BMI (<25 vs. ≥25 kg/m2) | 0.620 | | | | |
| preCEA (<5 vs. ≥5 ng/mL) | 0.169 | | | | |
| postCEA (<5 vs. ≥5 ng/mL) | 0.205 | | | | |
| Group (L vs.H) | 0.005 | 1.97 | 1.09–3.15 | 0.041 | |
| Tumor site (Right vs. Left) | 0.874 | | | | |
| Underlying disease | 0.030 | 1.65 | 0.77–3.84 | 0.218 | |
| Smoking | 0.539 | | | | |
| Alcohol | 0.257 | | | | |
| Complication | 0.367 | | | | |
| Histology (WD vs. MD) | 0.131 | | | | |

preCEA: preoperative carcinoembryonic antigen, WD: well differentiated, MD: moderately differentiated, HR: hazard ratio, 95% CI: 95% confidence interval

**Table 7. Uni- and multivariate analysis of risk factors associated with recurrence using Cox regression model in patients without diabetes.**

| Factors | Univariate | Multivariate | | |
|---|---|---|---|---|
| | *P* | HR | 95% CI | *P* |
| Age (<70 vs. ≥70 years) | 0.121 | | | |
| Gender (female vs. Male) | 0.709 | | | |
| BMI (<25 vs. ≥25 kg/m2) | 0.643 | | | |
| preCEA (<5 vs. ≥5 ng/mL) | 0.152 | | | |
| postCEA (<5 vs. ≥5 ng/mL) | 0.596 | | | |
| Group (L vs.H) | 0.021 | 2.25 | 1.13–4.51 | 0.021 |
| Tumor site (Right vs. Left) | 0.118 | | | |
| Underlying disease | 0.193 | | | |
| Smoking | 0.928 | | | |
| Alcohol | 0.996 | | | |
| Complication | 0.138 | | | |
| Histology (WD vs. MD) | 0.629 | | | |

preCEA: preoperative carcinoembryonic antigen, WD: well differentiated, MD: moderately differentiated, HR: hazard ratio, 95% CI: 95% confidence interval

factors for recurrence and mortality, but this did not produce statically meaningful results. Therefore, the ROC curve was used to determine the cut-off value between each of the preoperative and postoperative CEA levels and recurrence. Although the AUC was low, we were able to calculate a cut-off value of 3.305 ng/dL for preoperative CEA and 1.86 ng/dL for postoperative CEA. To overcome the low AUC, combinations of the two cut-offs were divided into three groups.

Our results showed that patients with higher perioperative CEA levels had a higher mean age. This is contrary to a paper published by Yanfeng Gao et al. [12], but was similar to a paper published by Tsuyoshi Konishi et al. [13] Smoking status in this study did not affect CEA levels, unlike in other studies [14, 15].

In this study, preoperative and postoperative CEA levels were classified into three patient groups. To achieve clearer results, groups from both extremes were included in the univariate

**Table 8. Uni- and multivariate analysis of risk factors associated with recurrence using Cox regression model in patients with diabetes.**

| Factors | Univariate | Multivariate | | |
|---|---|---|---|---|
| | *P* | HR | 95% CI | *P* |
| Age (<70 vs. ≥70 years) | 0.084 | 0.39 | 0.13–1.12 | 0.084 |
| Gender (female vs. Male) | 0.493 | | | |
| BMI (<25 vs. ≥25 kg/m2) | 0.169 | | | |
| preCEA (<5 vs. ≥5 ng/mL) | 0.867 | | | |
| postCEA (<5 vs. ≥5 ng/mL) | 0.778 | | | |
| Group (L vs.H) | 0.417 | | | |
| Tumor site (Right vs. Left) | 0.533 | | | |
| Underlying disease | - | | | |
| Smoking | 0.117 | | | |
| Alcohol | 0.912 | | | |
| Complication | 0.478 | | | |
| Histology (WD vs. MD) | 0.515 | | | |

preCEA: preoperative carcinoembryonic antigen, WD: well differentiated, MD: moderately differentiated, HR: hazard ratio, 95% CI: 95% confidence interval

and multivariate analyses to identify risk factors for recurrence and mortality. As a result, both recurrence and mortality showed significant results in group H compared with group L. As shown in Fig 1 on the extended line, comparing the Kaplan Meyer curve to determine disease-free survival and overall survival by group, group H shows a significant result compared with group L.

Using the univariate and multivariate models to identify risk factors for recurrence and mortality, both showed statistically significant results for group H compared with group L. In addition to mortality, age and risk factors also produced meaningful results, which is a natural result because the study included many elderly patients.

When designing this study, we thoroughly screened patients with stage llA colon cancer and excluded rectal cancer. According to the colon cancer part of the National Comprehensive Cancer Network guideline, if the pathologic stage is T3, N0, M0 and there are no high-risk features, the first choice of adjuvant treatment is observation, which is often used in clinical practice [4]. High-risk factors are defined as poorly differentiated / undifferentiated histology, lymphatic/vascular invasion, bowel obstruction, <12 lymph nodes examined, perineural invasion, localized perforation, or positive margins. For this reason, all patients with high-risk factors were excluded.

Adjuvant chemotherapy is known as the standard for stage III colon adenocarcinoma after resection. The addition of chemotherapy after surgical resection of stage III colon cancer provides a 22% to 32% advantage of overall survival (OS) and a 30% reduction in the relative risk of disease recurrence [16, 17]. Focusing on the obvious advantages of adjuvant chemotherapy in patients with stage III colon cancer, efforts were made to similarly treat patients with stage II colon cancer. As a result, it helps to lower survival and recurrence rates by identifying high-risk groups that can benefit from adjuvant treatment. However, current guidance does not support the use of CEA as an indicator for adjuvant chemotherapy [4, 18, 19]. However, there are several opinions on the relationship between postoperative CEA and prognosis. Several studies [20–22] have shown that postoperative CEA elevation is associated with prognosis in patients with stage II colon cancer, while another study [23] suggests that postoperative CEA levels in stage II disease do not affect disease-free survival. We also found no connection between postoperative CEA level and disease-free and overall survival.

We compared the serum levels of preoperative and postoperative CEA in this study to create groups for comparison. We confirmed that recurrence and overall survival were statistically significantly different between group H and group L based on the arbitrarily proposed cut-off value, although it was lower than the CEA reference value. The potential benefit of adjuvant chemotherapy in non-high-risk stage IIA colon cancer patients has not been fully evaluated. However, since the prognosis was confirmed to be poor in patients above the reference point suggested in this study, adjuvant chemotherapy should be considered for high-risk groups.

This analysis inevitably has the limitations and bias inherent in observational retrospective studies. For example, there is a difference in age between each group, which may be problematic because several studies report that there is a correlation between age and serum CEA level. The timing of preoperative and postoperative CEA measurement was not controlled. Although preoperative CEA was performed within 2 weeks before surgery, postoperative CEA was performed within 1 month after surgery. In most cases, the measurements of CEA level after surgery were confirmed by the examination conducted immediately before discharge, but when discharge was early, the results of the examination performed at the first outpatient follow-up were used. In addition, we have not controlled for other factors that can lead to false-positive elevated CEA levels, such as liver disease, gastritis, peptic ulcer disease, chronic obstructive

pulmonary disease, diverticulitis, and diabetes [13, 24]. In particular, diabetes was different in each group in this study, but it was not sufficiently controlled.

## Conclusions

Serum CEA level should be used as a predictor of recurrence or mortality after surgery in patients with low-risk stage IIA colon cancer. This study suggests that the recurrence rate and mortality rate are significantly higher when the preoperative CEA level is higher than 3.305 ng/dL and the postoperative CEA is higher than 1.86 ng/dL among patients with stage IIA colon cancer without high-risk features. Therefore, it is necessary to classify elevated CEA level as a high-risk feature, and adjuvant chemotherapy should also be considered.

## Supporting information

**S1 Data.**
(XLSX)

## Author Contributions

**Conceptualization:** Han-Gil Kim, Kang Young Lee, Hyuk Hur.

**Data curation:** Han-Gil Kim.

**Investigation:** Han-Gil Kim.

**Methodology:** Han-Gil Kim, Seung Yoon Yang, Yoon Dae Han, Min Soo Cho, Byung Soh Min, Kang Young Lee, Nam Kyu Kim.

**Project administration:** Seung Yoon Yang, Yoon Dae Han, Min Soo Cho, Nam Kyu Kim.

**Supervision:** Hyuk Hur.

**Validation:** Hyuk Hur.

**Writing – original draft:** Han-Gil Kim.

**Writing – review & editing:** Han-Gil Kim, Hyuk Hur.

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
