## [Decision Letter · Decision Letter 0]

30 Mar 2021

PONE-D-20-36945

Association of Perioperative Serum Carcinoembryonic Antigen Level and Recurrence in Low-risk Stage llA Colon Cancer

PLOS ONE

Dear Dr. Hur,

Thank you for submitting your manuscript to PLOS ONE. After careful consideration, we feel that it has merit but does not fully meet PLOS ONE’s publication criteria as it currently stands. Therefore, we invite you to submit a revised version of the manuscript that addresses the points raised during the review process.

We look forward to receiving your revised manuscript.

Kind regards,

Punita Dhawan

Academic Editor

PLOS ONE

Journal Requirements:

3. In the ethics statement in the manuscript and in the online submission form, please provide additional information about the patient records/samples used in your retrospective study, including: a) whether all data were fully anonymized before you accessed them; b) the date range (month and year) during which patients' medical records/samples were accessed; and c) the source of the medical records/samples analyzed in this work (e.g. hospital, institution or medical center name).

Reviewers' comments:

**Comments to the Author**

1. Is the manuscript technically sound, and do the data support the conclusions?

Reviewer #1: Yes

Reviewer #2: Partly

2. Has the statistical analysis been performed appropriately and rigorously? 

Reviewer #1: Yes

Reviewer #2: Yes

3. Have the authors made all data underlying the findings in their manuscript fully available?

Reviewer #1: Yes

Reviewer #2: Yes

4. Is the manuscript presented in an intelligible fashion and written in standard English?

Reviewer #1: Yes

Reviewer #2: Yes

5. Review Comments to the Author

Reviewer #1: References should be placed at the end of the sentences but before the 'period'. Also check the references for their contents and that must match with the information provided in the sentence. Also discuss role of adjuvant therapy with or without surgery in other stages of CRC.

Reviewer #2: The manuscript by Kim et al. reports the connection between perioperative serum carcinoembryonic antigen levels and recurrence in low-risk Stage llA colon cancer. However it has concerning discrepancies noted in major points. CEA expression is high already in Diabetic patient. So how we can say that CEA is remarkably associated with recurrence in low-risk Stage llA colon cancer. As a result, in order to draw a firm conclusion, authors should include a subgroups of both diabetic and non-diabetic patients in the H group.

Minor comments:

1.Abbreaviations should need to define when it has been coined first time in the manuscripts.

2 Minor changes to the manuscript's writing will also help to reinforce the paper.

6. PLOS authors have the option to publish the peer review history of their article (what does this mean?). If published, this will include your full peer review and any attached files.

Reviewer #1: No

Reviewer #2: No

---

## [Author Response · Author response to Decision Letter 0]

12 May 2021

Reviewer 1: I have incorporated all of your suggestions into my revision. It was a very useful suggestion. Thank you.

Reviewer 2: I have incorporated all of your suggestions into my revision. Thank you very much for your help

---

## [Editor Report · Decision Letter 1]

19 May 2021

Association of Perioperative Serum Carcinoembryonic Antigen Level and Recurrence in Low-risk Stage llA Colon Cancer

PONE-D-20-36945R1

Dear Dr. Hur,

We’re pleased to inform you that your manuscript has been judged scientifically suitable for publication and will be formally accepted for publication once it meets all outstanding technical requirements.

Kind regards,

Punita Dhawan

Academic Editor

PLOS ONE
---

## [Editor Report · Acceptance letter]

31 May 2021

PONE-D-20-36945R1 

Association of Perioperative Serum Carcinoembryonic Antigen Level and Recurrence in Low-risk Stage llA Colon Cancer 

Dear Dr. Hur:

I'm pleased to inform you that your manuscript has been deemed suitable for publication in PLOS ONE. Congratulations! Your manuscript is now with our production department. 

Kind regards, 

on behalf of

Dr. Punita Dhawan 

Academic Editor

PLOS ONE